# Geographic Variation in Preventable Hospitalizations among US Children with Autism

**DOI:** 10.3390/children10071228

**Published:** 2023-07-15

**Authors:** Wanqing Zhang, Khalilah R. Johnson

**Affiliations:** 1Department of Health Sciences, School of Medicine, University of North Carolina at Chapel Hill, Chapel Hill, NC 27599, USA; 2Division of Occupational Science and Occupational Therapy, School of Medicine, University of North Carolina at Chapel Hill, Chapel Hill, NC 27599, USA; khalilah_johnson@med.unc.edu

**Keywords:** ambulatory care sensitive conditions (ACSCs), autism, autistic children, preventable hospitalizations, racial differences, US census regions and divisions

## Abstract

There is a limited amount of research on geographic differences in preventable hospitalizations for ambulatory care sensitive conditions (ACSCs) among children with autism. The purpose of this study was to examine US regional differences in potentially preventable hospital admissions for pediatric inpatients diagnosed with autism. Hospital discharge data for six pediatric preventable conditions were obtained from the 2016–2019 National Inpatient Sample (NIS) under the US Agency for Healthcare Research and Quality. Geographic differences in preventable hospitalizations for children with autism were examined by US census regions and divisions. Multiple logistic regression analyses were conducted to examine child and clinical characteristics associated with ACSCs hospitalization across four US regions; the dependent variable was the likelihood of ACSCs hospitalization. Additionally, this study further explored the variation in preventable hospitalization among racial and ethnic groups for each region or division. Of the 138,305 autistic inpatients aged 2–17 years, about 10% had a primary diagnosis related to ACSCs. The results showed that the highest proportion of preventable hospitalizations for autistic children occurred in the middle Atlantic division of the northeast region. Racial differences were observed across all US regions, particularly in the northeast and south regions. Black children with autism were more likely to be hospitalized for ACSCs compared to White children with autism in three of the four US regions. Our results highlight the significant racial disparities in potentially avoidable hospitalizations among US children with autism. Examining geographic and racial differences in potentially avoidable hospitalizations could inform policy and practice while gaining a better understanding of pediatric patients with autism and where their families access health services. The findings of this study may help policymakers to identify where intervention is needed to tackle health inequities in the accessibility to quality primary care in the US. Further studies with more detailed investigation are recommended to better understand the mechanisms underlying these disparities, and to formulate effective regional policy and clinical practices while considering the unique needs and challenges of underserved children with autism.

## 1. Introduction

Autism is one of the pervasive developmental disorders, characterized by a range of conditions that affect communication, social interaction, and behavior [1,2]. The US Centers for Disease Control (2023) recently estimated that about 28 per 1000 eight-year-olds have an autism spectrum disorder (ASD) [3]. Children with ASD tend to have more co-occurring medical conditions, requiring costly and complex medical services such as emergency department visits and inpatient hospitalizations [4,5,6].

Preventable hospitalizations due to ambulatory care sensitive conditions (ACSCs) are inpatient admissions that could have been avoided through timely and effective management in the primary care setting [7,8]. For example, asthma is a chronic respiratory disease that can be managed properly with appropriate treatment and avoidance of asthma triggers in the outpatient setting. However, children with ASD from families of low socioeconomic status often lack access to services; thus, these children and their families may have greater stress related to unmet needs that could result in preventable hospitalizations for ACSCs [9,10,11]. The potentially avoidable hospitalizations add a substantial burden to these families and to the US healthcare system [12,13,14].

Previous studies indicate racial disparities in ASD. Racial minority children with ASD experienced limited access and greater barriers to receiving health services [15,16,17,18]. Black families were less likely to have ongoing primary care compared to White families [19,20], and Black children were less likely to receive family-centered care [21]. Disparities in access to timely health services for underserved minority children is a critical concern. Therefore, research examining racial disparities in preventable hospitalizations could identify barriers to accessing quality primary care services for disadvantaged children with autism, and hence help policymakers to formulate effective strategies and interventions.

Healthcare access and utilization among children with developmental problems or mental health issues differ widely across US regions [22,23]. Research has also indicated significant regional variability in ASD treatment. For example, a previous study reported that children from the Northeast region in the US had significantly higher specialty services utilization rate compared to children in other US regions [24]. Geographic disparities are important predictors for measuring healthcare access due to regional policies and other characteristics (e.g., healthcare market and service delivery) [25]. Research on geographic variation in preventable hospitalizations across US regions can provide data and insights on the effectiveness of preventive care interventions, access to care, as well as healthcare delivery systems. 

Studies of specific geographic areas enable us to examine the target population with similar characteristics; thus, this approach provides insights based on demographic and economic factors [26]. Examining regional differences could help identify specific socioeconomic and clinical factors that contribute to health disparities and inform the development of effective prevention and treatment strategies for vulnerable population groups. Data suggested that the proportion of preventable hospitalizations for ACSCs is higher in children with ASD compared with children without ASD [27]. However, little research has explored regional differences in potentially preventable hospitalizations in a pediatric population with autism. This analysis was based on US nationally representative hospital discharge data from the publicly available Nationwide Inpatient Sample (NIS) from the Healthcare Cost and Utilization Project (HCUP). We examined geographic patterns and disparities of preventable hospitalizations among children with autism, focusing on racial differences across US regions. Specifically, we used HCUP-NIS data to explore differences in preventable hospitalizations in US Census-defined geographic areas among pediatric inpatients with autism and analyze the difference in characteristics across regional cohorts. Our prior work has shown that the likelihood of preventable hospitalizations is significantly higher for Hispanic and Black children with autism compared with White children with autism [28]. This study further explores whether racial differences are associated with variation in preventable hospitalizations in each US census region or division. 

## 2. Materials and Methods

### 2.1. Data Source

The NIS is a part of the HCUP, which is a collection of healthcare databases developed by the Agency for Healthcare Research and Quality (AHRQ) in the US. The NIS includes hospital discharge data from all states participating in HCUP, representing approximately 97% of the US population [29]. The NIS contains information on patient demographics, primary and secondary diagnoses, hospital length of stay, total hospital charges, and other related information. Given the large sample size from the NIS, analyses can be conducted among special patient populations (e.g., patients with autism). The NIS provides a geographically diverse sample; it can be used to examine variations in the utilization and quality of health services for children with autism and associated comorbidities across different regions. In addition, discharge weights are provided for calculating national and regional estimates. We used multi-year aggregated datasets in this analysis to compare preventable hospitalizations among autistic children by US regions; a four-year pooled dataset was used to enable subgroup comparisons (e.g., racial subgroups in nine geographic divisions).

### 2.2. Participants and Measures

This analysis focused on pediatric patients with autism whose main reason for hospital admission was ACSCs-related diagnosis. The primary outcome measure was the hospital admission due to ACSCs; thus, our study sample includes pediatric inpatients ages 2 to 17 years with any secondary diagnoses of ASD. Based on the Pediatric Quality Indicator (PDI) from AHRQ-HCUP [30] and the results from our previous analysis using the NIS [28], preventable hospitalizations due to ACSCs in this current study included six PDIs. Among these PDIs, three ACSCs (i.e., asthma, constipation, and diabetes short-term complications) were categorized as chronic preventable conditions, while other three ACSCs (i.e., dehydration, gastroenteritis, and urinary infection) were categorized as acute preventable conditions. ASD and ACSCs were coded based on the ICD-10-CM (International Classification of Diseases, Tenth Revision, Clinical Modification). The term “ACSCs hospitalization” in this paper included any of these six conditions as the first-listed diagnosis in the hospital discharge record provided by the NIS. 

The US Census Bureau divides the country into four regions: northeast, midwest, south, and west. Each of these regions is then further divided into smaller geographic areas known as divisions [31]. These four census regions are grouped into nine US census divisions based on their geographic location: two divisions in the northeast (New England and middle Atlantic divisions), two divisions in the midwest (east north central and west north central), three divisions in the south (south Atlantic, east south central, and west south central), and two divisions in the west (mountain and pacific). Variables used in the regression analysis include patient and clinical characteristics. Race/ethnicity was classified into: non-Hispanic Whites, non-Hispanic Blacks, Hispanics, and other racial minorities (Asian and other races). Other child and clinical characteristics included: age (2–5 years, 6–12 years, and 13–17 years), sex (male or female), primary insurance payer type (private or public), resident location (urban or rural), hospital discharge disposition status (home discharge vs. others), and hospital length of stay. 

### 2.3. Statistical Analysis

ACSCs-related hospitalizations across US regions were examined using two measures: prevalence (proportion of ACSCs hospitalization during the four study years) and odds ratios (ORs) for ACSCs hospitalization. We applied multivariable logistic regression models to examine the different effects of the individual variables on the probability of ACSCs hospitalization among pediatric patients with ASD within four regional strata; these associations were estimated by calculating ORs and 95% confidence intervals (CIs). In addition, a separate division-level analysis was conducted in the northeast and south regions where significant racial differences were found. All differences between estimates noted in this analysis were statistically significant at the 0.05 level. Data were analyzed using SAS 9.4 Software (SAS Institute Inc., Cary, NC, USA). Institutional Review Board (IRB) approval was not required for this analysis as the NIS database does not include identifier information.

## 3. Results

This analysis included 138,305 hospitalizations among children with autism, and 14,110 of these pediatric hospitalizations (10.2%) were due to ACSCs during 2016–2019. Approximately 82.8% of autistic children hospitalized for ACSCs were in the age group of 2–12 years, with 74.7% of them being males. The racial/ethnic distribution of these child inpatients was as follows: 51% were non-Hispanic White, 19% were non-Hispanic Black, 21% were Hispanic, and 9% were of other races or ethnicities. Of the 14,110 autistic children hospitalized for ACSCs, 8525 (60.4%) were for chronic ACSCs and 5585 (39.6%) were for acute ACSCs. Among chronic ACSCs, 49.4% (N = 4210) had a primary diagnosis of asthma.

Figure 1 shows the prevalence in pediatric hospitalizations due to ACSCs for children with autism in nine US divisions. The prevalence of all ACSCs ranged from 6.33% (west north central) to 12.69% (middle Atlantic). The prevalence of chronic ACSCs ranged from 3.38% (west north central) to 8.09% (middle Atlantic), while the prevalence of acute ACSCs ranged from 2.96% (west north central) to 5.26% (west south central). The prevalence varied by four census regions for all ACSCs hospitalizations and ranged from 8.06% in the midwest to 12.15% in the northeast (*p* < 0.001); chronic ACSCs ranged from 4.95% in the midwest to 7.69% in the northeast (*p* < 0.001); and acute ACSCs ranged from 3.11% in the midwest to 4.46% in the northeast (*p* < 0.001). Asthma was the most frequent reason for ACSCs-related hospitalizations in all four regions, with the prevalence ranging from 2.06% (midwest) to 5.08% (northeast).

Table 1, Table 2, Table 3 and Table 4 present the multivariable analysis results from four US census regions among pediatric inpatients with autism. Some characteristics related to ACSCs hospitalization for these patients were shared across the four regions. Across all regions, children younger than six years were more likely to be hospitalized for ACSCs than those aged 6–17; hospital lengths of stay were significantly shorter for ACSCs hospitalization vs. non-ACSCs hospitalization. Except for the midwest, publicly insured children with autism were more likely to be hospitalized for ACSCs than privately insured children with autism in the three other regions. In the northeast and south regions, children hospitalized for ACSCs were more likely to be discharged routinely compared to their non-ASD counterparts after holding other variables constant.

The adjusted regression analysis reveals significant regional variation in the association between race and ACSCs hospitalization among pediatric inpatients with autism. In general, Black children with autism were more likely to be hospitalized for ACSCs compared to White children with autism in the same regions. Black children with autism in three of the four regions (i.e., northeast, midwest, and south) were more likely to be hospitalized for chronic ACSCs than White children with autism in the same regions (Table 1, Table 2 and Table 3). In contrast, Black children with autism were less likely than White children with autism to be hospitalized for acute ACSCs in the northeast (adjusted OR 0.547, 95% CI, 0.354–0.845) and in the south (OR 0.648, 95% CI, 0.472–0.890). In addition, racial differences were evident for asthma hospitalization across all four regions. Comparing non-Hispanic Blacks to Whites, the OR for asthma hospitalization was highest in the south region (OR 5.101, 95% CI, 3.704–7.026), followed by the northeast (OR 4.790, 95% CI, 3.375–6.800), midwest (OR 3.569, 95% CI, 2.370–5.376), and west (OR 2.264, 95% CI, 1.259–4.071) region.

Within the northeast region, Hispanic children with autism were more likely than White children with autism to be hospitalized for ACSCs in the mid-Atlantic division (adjusted OR 1.352, 95% CI, 1.059–1.725); there were no statistically significant differences found in the New England division between these two racial groups. Within the south region, Black children with autism were more likely than were White children with autism to be hospitalized for ACSCs in the south Atlantic division (adjusted OR 1.436, 95% CI, 1.131–1.825); however, there were no statistically significant Black-White differences in the east south central and west south central divisions.

## 4. Discussion

Findings from this analysis highlight the geographic differences that exist in ACSCs-related hospitalizations for children with autism. The northeast region, particularly in the middle Atlantic division, had a higher prevalence of ACSCs hospitalization in our study population when compared to most other regions. This finding is consistent with a previous study on ASD hospitalization for both children and adults in the US, where ASD was the principal diagnosis for hospital admission [32]. The result from this analysis indicates that the patterns of ACSCs hospitalization for children with autism differ in both demographic and clinical characteristics across US Census Bureau-defined regions and divisions. Child and clinical factors such as age, race, health insurance type, and hospital discharge status were associated with these differences. Children with autism from low-income families have problems with getting timely access to care [33]. Our recent study reveals racial/ethnic and income disparities in potentially avoidable hospitalizations among children with autism [28]. More specifically, a Texas study reported that geographic disparities in pediatric hospitalization for asthma were significantly influenced by socioeconomic inequities related to health insurance and race/ethnicity in Lubbock County [34]. Notably, our multivariable analysis results show that race is a unique factor associated with potentially preventable hospitalizations for children with autism across geographic regions and divisions in the US. 

Our analysis suggests that Black children with autism are overrepresented for ACSCs hospitalization in the 3 of the 4 US regions. In addition, racial/ethnic disparities in ACSCs hospitalization were observed within the northeast and south regions. For example, Black children with autism had a greater likelihood of ACSCs hospitalization in the south Atlantic division, while no Black–White difference was found in both the east south central and west south central divisions. Variations in socioeconomic status among different divisions within the south region may play a role. The south Atlantic division includes counties and areas with higher levels of social-economic and educational disparities, which may contribute to poorer access to healthcare services among underserved children and their families [35]. Differences in the population composition by racial and ethnic groups across these divisions could influence the findings. Metropolitan cities such as Atlanta and Baltimore in the south Atlantic division have a high proportion of Black residents [36], which could contribute to the observed disparity. For example, a study pointed out that a higher proportion of Black children with asthma resided in the south Atlantic region [37]. In addition, cultural beliefs and healthcare-seeking behaviors may also influence preventable hospitalizations among families with autistic children. Differences in caregiver knowledge about autism, and access to support systems can influence healthcare-seeking behaviors and health services utilization. Children with co-occurring preventable conditions and ASD require a high level of continuity of care and care coordination between their families and healthcare providers, as they may need specialized preventive care to avoid hospital admissions due to cognitive or physical impairments.

Asthma is a common chronic condition among children with autism [38,39]. This analysis shows that asthma is the most prevalent condition among autistic children across all regions in the US. Autistic children face unique challenges in managing their asthma, including difficulties in communication and understanding their symptoms, as well as a higher risk for sensory sensitivities that can trigger asthma exacerbations. Children with autism may also face additional stressors, such as social isolation and behavioral issues, which can further impact the management of their asthma. Furthermore, White–Black disparity in preventable asthma hospitalization is evident. Black children with autism are about two to five times more likely than White children with autism to have inpatient admissions for asthma in all US regions. Non-Hispanic Blacks tended to receive less education and information about asthma medications and other asthma management options [40]. This finding highlights the need for increased resources and attention to address the disparity in potentially preventable hospitalizations for asthma among Black children with autism. Further research is needed to understand the underlying causes of this disparity, in terms of access to healthcare, availability of resources for managing asthma, and social determinants of health.

In this study, we were able to explore regional differences in ACSCs hospitalization using the stratified systematic sample of hospital discharge datasets provided by the HCUP-NIS. This report provides insights into potential disparities and differences in potentially avoidable hospitalizations among autistic children across US regions. However, the current analysis has several limitations. First, the NIS only includes data from a subset of hospitals in the US, which can limit its ability to accurately represent geographic differences in hospital utilization. Second, we could not identify a specific US state within a US division as state-level data were unavailable in the publicly available NIS datasets. Third, the NIS is cross-sectional, so we only make geographic comparisons at a single point in time. Fourth, the quality and consistency of diagnostic coding for ACSCs in the NIS may vary across US hospitals. This can lead to potential misclassifications. In addition, the NIS lacks detailed information about other racial/ethnic groups (e.g., multiracial individuals); our analyses combined other race and ethnicity groups (e.g., Asian or Pacific Islander, Native American, and others) because of small sample sizes. Finally, the NIS only includes discharge-level data, rather than individual-level records, which can limit the ability to conduct detailed analyses on specific patient characteristics, outcomes, or follow-up information. Despite these limitations, our comparative study of ACSCs hospitalization across the geographic and racial spectrum helps to inform the development of relevant policy and practices that can effectively address access problems for timely and quality health services among underserved children with autism.

## 5. Conclusions

Significant regional and racial differences were found in ACSCs-related hospitalizations among children with autism in the US. These differences are influenced by a combination of various factors in each geographic region, including age, race, health insurance type, and clinical characteristics such as hospital length of stay. Our findings can have potential implications for the reduction of preventable hospitalizations across US regions, as each geographic area has unique characteristics that may contribute to differences in the likelihood of potentially avoidable hospitalizations for children with autism. The results of this analysis may provide direction for further research to understand the specific dynamics within a geographic division, and hence inform targeted interventions aimed at reducing disparities and promoting health equity for vulnerable children with autism and other developmental disabilities across US regions.

## Figures and Tables

**Figure 1 children-10-01228-f001:**
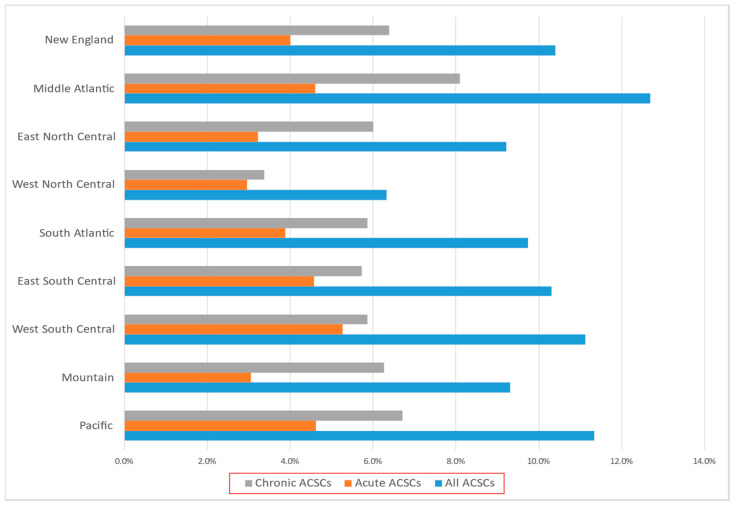
Prevalence in pediatric hospitalizations due to ACSCs among pediatric inpatients with autism across nine US divisions.

**Table 1 children-10-01228-t001:** Multivariable regression results for ACSCs hospitalization among children with autism in the Northeast.

Variable	All ACSCs	Acute ACSCs	Chronic ACSCs	Asthma
	OR (*p* Value)	OR (*p* Value)	OR (*p* Value)	OR (*p* Value)
**Race**				
Black (vs. White)	1.509 (0.0004)	0.547 (0.0065)	2.345 (<0.0001)	4.790 (<0.0001)
Hispanic (vs. White)	1.332 (0.0104)	1.094 (0.5861)	1.473 (0.0056)	2.998 (<0.0001)
**Sex**				
Female (vs. male)	0.975 (0.7986)	1.002 (0.9873)	0.961 (0.7404)	0.947 (0.7176)
**Age**				
6–12 (vs. 2–5)	0.595 (<0.0001)	0.427 (<0.0001)	0.792 (0.0394)	0.585 (<0.0001)
13–17 (vs. 2–5)	0.263 (<0.0001)	0.261 (<0.0001)	0.303 (<0.0001)	0.113 (<0.0001)
**Insurance Type**				
Public (vs. private)	1.201 (0.0451)	0.955 (0.7387)	1.361 (0.0069)	1.115 (0.4405)
**Rural-urban Areas**				
Rural (vs. urban)	0.638 (0.0540)	0.356 (0.0243)	0.902 (0.7013)	0.848 (0.6958)
**Discharge Status**				
Routine (vs. others)	1.493 (0.0342)	1.227 (0.4707)	1.619 (0.0450)	1.432 (0.2497)
**Hospital Days**				
Length of Stay	0.873 (<0.0001)	0.897 (<0.0001)	0.867 (<0.0001)	0.802 (<0.0001)

**Table 2 children-10-01228-t002:** Multivariable regression results for ACSCs hospitalization among children with autism in the Midwest.

Variable	All ACSCs	Acute ACSCs	Chronic ACSCs	Asthma
	OR (*p* Value)	OR (*p* Value)	OR (*p* Value)	OR (*p* Value)
**Race**				
Black (vs. White)	1.790 (<0.0001)	1.127 (0.5690)	2.182 (<0.0001)	3.569 (<0.0001)
Hispanic (vs. White)	1.178 (0.4096)	0.928 (0.8172)	1.355 (0.2099)	1.764 (0.1091)
**Sex**				
Female (vs. male)	1.048 (0.6765)	1.714 (0.0007)	0.704 (0.0235)	0.482 (0.0063)
**Age**				
6–12 (vs. 2–5)	0.535 (<0.0001)	0.420 (<0.0001)	0.701 (0.0154)	0.370 (<0.0001)
13–17 (vs. 2–5)	0.240 (<0.0001)	0.140 (<0.0001)	0.386 (<0.0001)	0.054 (<0.0001)
**Insurance Type**				
Public (vs. private)	0.967 (0.7420)	0.942 (0.6943)	0.992 (0.9471)	1.189 (0.4059)
**Rural-urban Areas**				
Rural (vs. urban)	0.982 (0.8895)	1.019 (0.9235)	0.956 (0.7915)	0.589 (0.1016)
**Discharge Status**				
Routine (vs. others)	1.263 (0.3480)	0.825 (0.5574)	1.761 (0.1054)	4.786 (0.1231)
**Hospital Days**				
Length of Stay	0.860 (<0.0001)	0.911 (0.0054)	0.830 (0.0002)	0.836 (0.0945)

**Table 3 children-10-01228-t003:** Multivariable regression results for ACSCs hospitalization among children with autism in the South.

Variable	All ACSCs	Acute ACSCs	Chronic ACSCs	Asthma
	OR (*p* Value)	OR (*p* Value)	OR (*p* Value)	OR (*p* Value)
**Race**				
Black (vs. White)	1.278 (0.0077)	0.648 (0.0074)	1.856 (<0.0001)	5.101 (<0.0001)
Hispanic (vs. White)	0.930 (0.4756)	0.893 (0.4314)	0.978 (0.8640)	2.095 (0.0002)
**Sex**				
Female (vs. male)	1.033 (0.6920)	1.493 (0.0003)	0.729 (0.0055)	0.521 (0.0006)
**Age**				
6–12 (vs. 2–5)	0.690 (<0.0001)	0.504 (<0.0001)	0.974 (0.8188)	0.717 (0.0296)
13–17 (vs. 2–5)	0.304 (<0.0001)	0.228 (<0.0001)	0.448 (<0.0001)	0.160 (<0.0001)
**Insurance Type**				
Public (vs. private)	1.192 (0.0254)	0.878 (0.2318)	1.544 (<0.0001)	1.768 (0.0014)
**Rural-urban Areas**				
Rural (vs. urban)	1.026 (0.8056)	0.356 (0.0243)	1.128 (0.3538)	1.029 (0.8916)
**Discharge Status**				
Routine (vs. others)	1.773 (0.0049)	0.967 (0.8904)	3.624 (0.0003)	1.998 (0.1296)
**Hospital Days**				
Length of Stay	0.798 (<0.0001)	0.836 (<0.0001)	0.782 (<0.0001)	0.714 (<0.0001)

**Table 4 children-10-01228-t004:** Multivariable regression results for ACSCs hospitalization among children with autism in the West.

Variable	All ACSCs	Acute ACSCs	Chronic ACSCs	Asthma
	OR (*p* Value)	OR (*p* Value)	OR (*p* Value)	OR (*p* Value)
**Race**				
Black (vs. White)	1.256 (0.2280)	0.991 (0.9783)	1.390 (0.1389)	2.264 (0.0063)
Hispanic (vs. White)	1.029 (0.8009)	1.219 (0.2427)	0.914 (0.5245)	1.109 (0.6232)
**Sex**				
Female (vs. male)	1.117 (0.3011)	1.372 (0.0469)	0.958 (0.7545)	0.609 (0.0292)
**Age**				
6–12 (vs. 2–5)	0.717 (0.0020)	0.578 (0.0007)	0.874 (0.3195)	0.541 (0.0008)
13–17 (vs. 2–5)	0.316 (<0.0001)	0.224 (<0.0001)	0.434 (<0.0001)	0.197 (<0.0001)
**Insurance Type**				
Public (vs. private)	1.367 (0.0038)	1.301 (0.1193)	1.366 (0.0196)	1.620 (0.0168)
**Rural-urban Areas**				
Rural (vs. urban)	1.212 (0.4022)	1.036 (0.9272)	1.294 (0.3333)	0.761 (0.6092)
**Discharge Status**				
Routine (vs. others)	1.268 (0.3511)	1.687 (0.2531)	1.078 (0.7997)	2.637 (0.1795)
**Hospital Days**				
Length of Stay	0.875 (<0.0001)	0.901 (0.0023)	0.867 (<0.0001)	0.730 (<0.0001)

## Data Availability

The data that support the findings of this study are available from the corresponding author upon reasonable request.

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
