# Peer review of "Geographic Variation in Preventable Hospitalizations among US Children with Autism"

_children, 2023, doi:10.3390/children10071228_

Round 1

Reviewer 1 Report

Please refer to the attached pdf for the detailed comments. Thank you 

Author Response

Response to Reviewer 1

Lines 12

What is the dependent and independent variables?

Authors’ Response: Multiple logistic regression analyses were conducted to examine child and clinical characteristics associated with ACSCs hospitalization across 4 US regions; the dependent variable was the likelihood of ACSCs hospitalization (new lines 16-19). Please also see detailed description in the Methods section (new lines 128-134).

Lines 21-22

What does this statement means? please rephrase.

Authors’ Response: This sentence was rephrased as per the reviewer’s suggestion (new lines 24-25).

Lines 23-24

How can this info implicate the policy? management? improve the prevention of hospitalization of children with autism in the US.

Authors’ Response: Examining geographic and racial differences in potentially avoidable hospitalizations could inform policy and practice while gaining better understanding of the pediatric patients with autism and where their families access health services. The findings of this study may help policy makers to identify where intervention is needed to tackle health inequities in the accessibility to quality primary care in the US. We added this statement to the abstract (new lines 27-31).

Lines 84

Author need to explicitly mention all the variables that the study is assessing in correlation to the racial differences here as discussed in the MnM. The aim here need to be consistent with the aim mentioned in the abstract.

Authors’ Response: The variables in the analysis included demographic and clinical variables such as child age and sex, insurance payer type, resident location, hospital discharge status, and hospital length of stay (new lines 128-134). As per the reviewer’s comment, we rephrased the purpose statement in the abstract to keep consistency.

Lines 103-104

Is there are reason why children primarily diagnosed with ASD is excluded? or what do u mean by secondary ASD diagnoses?

Authors’ Response: This analysis focused on pediatric patients with autism whose main reason for hospital admission was ACSCs-related diagnosis. The primary outcome measure was the hospital admission due to ACSCs; thus, our study sample includes pediatric inpatients ages 2 to 17 years with any secondary diagnoses of ASD. Clarity was added (new lines 109-112).

Figures

is this Northeast as used in the writing? please just use one term. comment as below? is this Midwest? i suggest to use the same term through our manuscript as this will help non American who are not quite familiar with the terms of the division.

Authors’ Response: These 4 census regions are grouped into 9 US census divisions based on their geographic location: 2 divisions in the Northeast (New England and Middle Atlantic divisions), 2 divisions in the Midwest (East North Central and West North Central), 3 divisions in the South (South Atlantic, East South Central, and West South Central), and 2 divisions in the West (Mountain, and Pacific). Clarity was added (new lines 124-128).

Lines 160, 161 etc.

Children with autism , children with ASD. Please improve through out the manuscript. thank u

Authors’ Response: We have made the edits as per the reviewer’s suggestion throughout the manuscript.

Lines 272-288 (Conclusion section).

In my opinion can be simplified and focused on the main findings in correlation to the objective, implication of the findings, recommendation and future research.

Authors’ Response: As per the reviewer’s suggestion, the conclusion section was revised/simplified.

Reviewer 2 Report

I would like you to drill down to explain more about how race/ethnicity data is collected for the analysis. I suspected that the 9% indicating other may be either individuals of multiple races and/or individuals who fear designating their race. I feel that this has broad implications for your conclusions especially if you are finding that race/ethnicity is not self-reported by participants or caregivers. I know that my institution has taken additional lengths to explore how race data is collected in research projects. Methodology is really important. If you are finding that this is not clear for NIS, this should be added as an additional limitation of the current data.

Author Response

Reviewer 2

I would like you to drill down to explain more about how race/ethnicity data is collected for the analysis. I suspected that the 9% indicating other may be either individuals of multiple races and/or individuals who fear designating their race. I feel that this has broad implications for your conclusions especially if you are finding that race/ethnicity is not self-reported by participants or caregivers. I know that my institution has taken additional lengths to explore how race data is collected in research projects. Methodology is really important. If you are finding that this is not clear for NIS, this should be added as an additional limitation of the current data.

Authors’ Response: The NIS lacks detailed information about other racial/ethnic groups such as multiracial individuals. This was added as an additional limitation (new lines 277-280).